# Targeted Unlearning with Single Layer Unlearning Gradient

**Zikui Cai** [1][*]
zikui@umd.edu

**Yaoteng Tan** [2]
ytan082@ucr.edu

**M. Salman Asif** [2]
sasif@ucr.edu

[1]University of Maryland, College Park, MD
[2]University of California, Riverside, CA

## Abstract

The unauthorized generation of privacy-related and copyright-infringing content using generative-AI is becoming a significant concern for society, raising ethical, legal, and privacy issues that demand urgent attention. Recently, machine unlearning techniques have arisen that attempt to eliminate the influence of sensitive content used during model training, but they often require extensive updates in the model, reduce the utility of the models for unrelated content, and/or incur substantial computational costs. In this work, we propose a novel and efficient method called Single Layer Unlearning Gradient (SLUG), that can unlearn targeted information by updating a single targeted layer of a model using a one-time gradient computation. We introduce two metrics: layer importance and gradient alignment, to identify the appropriate layers for unlearning targeted information. Our method is highly modular and enables selective removal of multiple concepts from the generated outputs of widely used foundation models (e.g., CLIP), generative models (e.g., Stable Diffusion) and Vision-Language models. Our method shows effectiveness on a broad spectrum of concepts ranging from concrete (e.g., celebrity name, intellectual property figure, and object) to abstract (e.g., novel concept and artistic style). Our code is available at https://github.com/CSIPlab/SLUG.

## 1 Introduction

Modern generative models, including large language models (LLMs) [2, 22], Stable Diffusion (SD) [32, 39], and vision language mdoels (VLMs) [43, 25] leverage vast amounts of data for training. While these large unsupervised datasets enhance performance under scaling law [17], they also raise serious data privacy and legal compliance [1, 34] concerns. Completely abandoning trained model weights and re-training large models from scratch using scrutinized dataset is prohibitively expensive, highlighting the need for efficient unlearning methods.

Machine unlearning (MU) [5, 29] refers to a set of techniques designed to reverse the learning process, which aims to efficiently remove targeted information from a trained model without re-training the model from scratch. MU has three main objectives: **(1) Low computational cost**, as the naïve approach of re-training models usually achieves the best result (exact unlearning) at the expense of large computational cost. **(2) Effective unlearning**, to ensure that the model forgets the intended data completely. **(3) Utility retention**, maintaining

---

[*]This work was done while the author was at the University of California, Riverside.

38th Conference on Neural Information Processing Systems Workshop (NeurIPS 2024W).

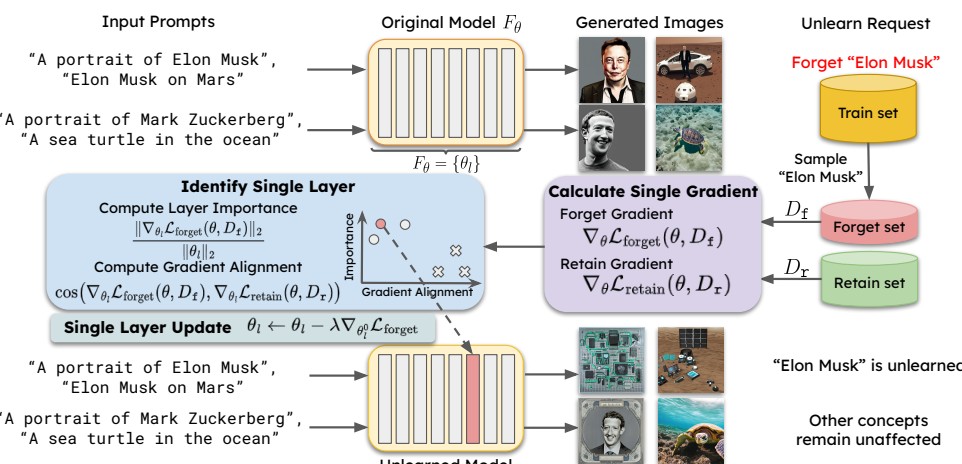

**Figure 1:** Overview of our proposed **S**ingle **L**ayer **U**nlearning **G**radient (SLUG) framework. Given an unlearning query, we curate a forget set and retain set, then compute corresponding gradients. The gradient alignment guide identifying single layer updates for unlearning. A binary search helps determine the step size $\lambda$, effectively erasing specified concepts while preserving the model's utility.

the original model performance, in terms of accuracy and utility on data/tasks unrelated to the intended forgotten data. Current MU methods often fall short of meeting all these objectives simultaneously. Traditional methods like fine-tuning (FT) [36] and gradient ascent (GA) [35] struggle to balance effective forgetting with utility preservation. More recent techniques such as saliency unlearning (SalUn) [8] and selective synaptic dampening (SSD) [9] attempt to address this by identifying and updating only salient parameters. While these methods improve overall unlearning performance, they still face the following challenges: (1) they usually involve iterative updates over the model parameters, resulting in high computational costs [8]. (2) The significant weights targeted for updates are often spread throughout the model, offering limited insight into the model's structure. (3) They require careful hyperparameter tuning, including learning rate, number of iterations, and parameters for selecting suitable masks in saliency approaches.

In this work, we propose a novel and efficient method for unlearning targeted information, namely **Single Layer Unlearning Gradient (SLUG)**. Figure 1 provides an overview of our proposed framework. SLUG performs three main steps using given unlearning query with retain and forget sets: (1) calculate one-time gradients for the forget and retain losses; (2) identify a single layer with high importance to the forget set and low relevance to the retain set; (3) update the targeted layer along a linear path using one-time calculated gradient. In addition to its efficiency and effectiveness, our methods offers higher modularity and better interpretability compared to [8, 9]. SLUG precisely identifies layers associated with distinct concepts, which provides insights into the features learned by different layers and their functionalities, offering generalized guidance for new tasks and model architectures design.

## 2 Background

### 2.1 Machine unlearning preliminaries

The goal of MU is to remove the influence of a specific subset of training data, $D_{\mathtt{f}} \subset D$, on a pre-trained model $F_\theta(D)$ with parameters $\theta$. The challenge is to make this process more efficient than re-training the model on the retain set $D_{\mathtt{r}} = D \setminus D_{\mathtt{f}}$. The unlearning algorithm $U$ should produce an unlearned model $F_{\theta_{\mathtt{f}}} = U(F_\theta(D), D, D_{\mathtt{f}})$ that is functionally equivalent to a model retrained only on $D_{\mathtt{r}}$ (i.e., $F_{\theta_{\mathtt{r}}}(D_{\mathtt{r}})$), which can be formalized as:

$$\min_\theta \underbrace{\frac{1}{N_{\mathtt{r}}} \sum_{(x_{\mathtt{r}}, y_{\mathtt{r}}) \in D_{\mathtt{r}}} \ell(F_\theta(x_{\mathtt{r}}), y_{\mathtt{r}})}_{\mathcal{L}_{\text{retain}}} - \underbrace{\frac{\alpha}{N_{\mathtt{f}}} \sum_{(x_{\mathtt{f}}, y_{\mathtt{f}}) \in D_{\mathtt{f}}} \ell(F_\theta(x_{\mathtt{f}}), y_{\mathtt{f}})}_{\mathcal{L}_{\text{forget}}} \tag{1}$$

where $N$ is the number of elements in $D$, $\alpha$ is a balancing factor, and $\ell$ is the loss function.

### 2.1.1 Vision language alignment

Traditional MU approaches struggle with high computational costs and limited scalability, which restricts their application to small-scale image classification models [16, 9]. In contrast, our method breaks away from these constraints by offering superior scalability and flexibility that is suitable for large multi-modal foundation models such as CLIP, SD, and VLMs.

CLIP [31], in particular, is pivotal in advancing multi-modal models by aligning visual and textual representations through contrastive loss [7]:

$$\ell = \frac{1}{2N} \sum_{i=1}^{N} \left( \ell_{i2t}(i) + \ell_{t2i}(i) \right), \tag{2}$$

$$\ell_{i2t}(i) = -\log \frac{\exp(\cos(\mathbf{v}_i, \mathbf{t}_i)/\tau)}{\sum_{j=1}^{N} \exp(\cos(\mathbf{v}_i, \mathbf{t}_j)/\tau)}, \quad \ell_{t2i}(i) = -\log \frac{\exp(\cos(\mathbf{t}_i, \mathbf{v}_i)/\tau)}{\sum_{j=1}^{N} \exp(\cos(\mathbf{t}_i, \mathbf{v}_j)/\tau)}. \tag{3}$$

Here, $\mathbf{v}_i$ is the normalized image embedding from the vision model $f_v$, and $\mathbf{t}_i$ is the normalized text embedding from the text model $f_t$. The temperature $\tau$ controls the sharpness of the softmax probability distribution, while cosine similarity is defined as $\cos(\mathbf{v}_i, \mathbf{t}_j) = \mathbf{v}_i \cdot \mathbf{t}_j$. Minimizing this contrastive loss aligns the vision and language representations in the embedding space. In unlearning, our goal is to break these learned alignments.

### 2.1.2 Loss functions for gradient calculation

Selection of an appropriate loss functions to perform unlearning is critical. In the scenario for contrastive learning we focus on contrastive loss. The loss for retain and forget sets are defined as follows:

$$\mathcal{L}_{\text{retain}} = \frac{1}{2N} \sum_{i=1}^{N} \left( \ell_{i2t}(i) + \ell_{t2i}(i) \right), \quad \mathcal{L}_{\text{forget}}(\mathbf{v}_i, \mathbf{t}_j) = 1 - \cos(\mathbf{v}_i, \mathbf{t}_j) \tag{4}$$

Retain loss is the original contrastive loss as in equation 2. For the forget loss, we employ the cosine embedding loss that directly pushes the embeddings of positive pairs away while not tampering with the embeddings of negative pairs. Using the original contrastive loss as forget loss will result in ineffective unlearning.

## 3 Single layer unlearning gradient (SLUG)

Our approach improves the state-of-the-art along three axes: (1) low computational cost, (2) effective unlearning, and (3) high retention of general utility. The framework is illustrated in Figure 1.

### 3.1 Layer identification

SLUG is inspired by the nature of different layers in deep networks learn distinct features [41, 30, 12]. To efficiently unlearn, our goal is to identify the layers most critical to unlearn targeted concepts while preserving the model's functionality. To achieve this, we perform unlearning within the "nullspace" of the retain set, focusing on layers that minimally impact retained data performance while effectively removing the targeted features.

To measure the influence of each parameter, similar to [3, 9], we use the Fisher information matrix[18, 14, 19], approximated by its diagonal for computational feasibility:

$$\mathcal{I}_D(\theta) = -\mathbb{E}\left[ \frac{\partial^2}{\partial \theta^2} \log L(\theta; D) \right] = \mathbb{E}\left[ \left( \frac{\partial}{\partial \theta} \log L(\theta; D) \right) \left( \frac{\partial}{\partial \theta} \log L(\theta; D) \right)^{\mathsf{T}} \right] \tag{5}$$

The diagonal elements reflect the sensitivity of the log-likelihood to parameter changes, and we extend this to layers by aggregating sensitivities. The importance of a layer is

determined by the ratio of the $\ell_2$ norm of the forget loss gradients to the $\ell_2$ norm of the layer's parameters:

$$\texttt{Importance of layer l:}\quad \text{Importance}(l) = \frac{\sqrt{\mathcal{I}_{D_{\mathtt{f}}}(\theta_l)}}{\|\theta_l\|_2} = \frac{\|\nabla_{\theta_l}\mathcal{L}_{\text{forget}}(\theta, D_{\mathtt{f}})\|_2}{\|\theta_l\|_2} \quad (6)$$

Importance of layer alone is insufficient. We also ensure that forget gradients are nearly orthogonal to retain gradients by minimizing the gradient alignment:

$$\texttt{Gradient alignment:}\quad \text{Alignment}(l) = \cos\big(\nabla_{\theta_l}\mathcal{L}_{\text{forget}}(\theta, D_{\mathtt{f}}), \nabla_{\theta_l}\mathcal{L}_{\text{retain}}(\theta, D_{\mathtt{r}})\big) \quad (7)$$

Small alignment would prevent unlearning updates from negatively affecting the retain set. To balance both objectives, we use the concept of a Pareto optimal set [28], optimizing both importance and gradient orthogonality. Figure 2 illustrates the Pareto front for unlearning a person identity from CLIP ViT-B-32, showing layers that unlearn the forget set without harming retain data.

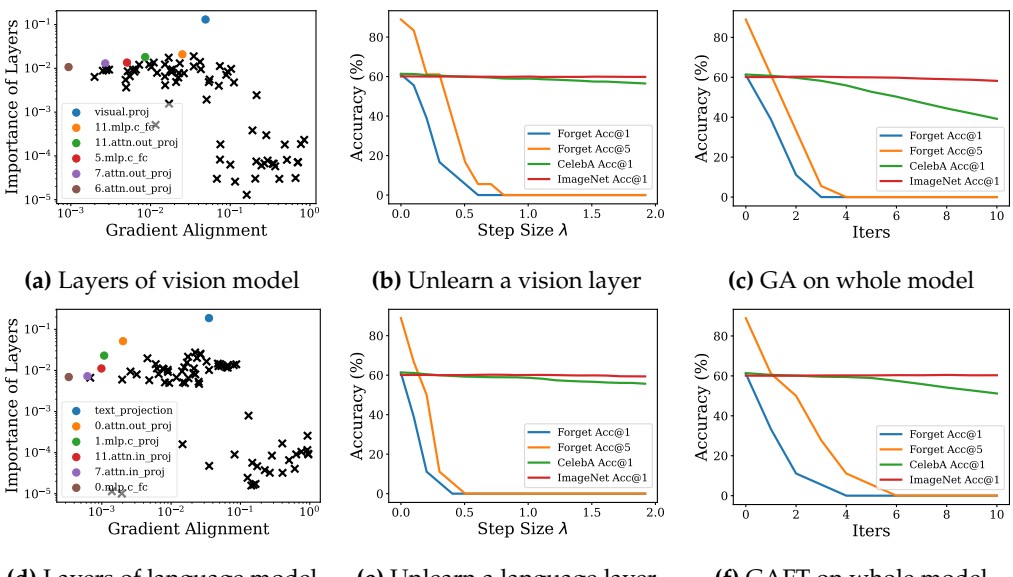

**(a)** Layers of vision model     **(b)** Unlearn a vision layer     **(c)** GA on whole model

**(d)** Layers of language model     **(e)** Unlearn a language layer     **(f)** GAFT on whole model

**Figure 2:** Layer identification (a,d) and unlearning with a single gradient (b,e). The first column shows gradient alignment and importance metrics for vision and language models from CLIP ViT-B-32, highlighting layers on the Pareto front for unlearning an identity. The second column demonstrates effective unlearning by updating identified layers along a single gradient direction without significantly impacting retain set performance. The third column shows that iterative methods (GA and GAFT) offer no advantage over a single gradient and require early stopping to prevent over-unlearning.

## 3.2 Linearizing unlearning trajectory

Existing unlearning methods calculate gradients at each iteration to update model parameters, which significantly increases computational complexity. However, inspired by task arithmetic [15] and the linear nature of many optimization problems [21], we observe that repeated gradient calculations may be redundant. Instead, we propose calculating the gradient only once for the initialized model and updating the parameters $\theta_l$ of any layer $l$ in a weight-arithmetic fashion. Specifically, the weights are updated along a fixed gradient direction in every iteration:

$$\theta_l^{(t)} \leftarrow \theta_l^{(0)} - \lambda^{(t)}\nabla_{\theta_l}\mathcal{L}_{\text{forget}}(\theta, D_{\mathtt{f}})\Big|_{\theta=\theta^{(0)}} \quad (8)$$

Here, $\theta_l^{(t)}$ represents the parameters of layer $l$ at iteration $t$, with $\theta_l^{(0)}$ being the initial parameters. The gradient $\nabla_{\theta_l} \mathcal{L}_{\text{forget}}(\theta, D_{\mathtt{f}})\big|_{\theta=\theta^{(0)}}$ is calculated once, based on the forget loss $\mathcal{L}_{\text{forget}}$ evaluated on the forget set $D_{\mathtt{f}}$. The step size $\lambda^{(t)}$ controls the update magnitude.

Instead of recalculating gradients, we use the initial gradient direction for updates, effectively linearizing the unlearning trajectory. This approach reduces computational complexity yet ensuring effective unlearning. To search a proper step size $\lambda^{(t)}$, we perform binary search along the linearized path, halting when the evaluation metric indicates satisfactory unlearning without harming performance on the retain set. This method strikes a balance between efficiency and precision, maintaining model utility while achieving unlearning goals.

### 3.3 Generalization to Stable Diffusion and VLMs

By harnessing effective unlearning in CLIP models, SLUG can be extended to larger generative models built on CLIP, such as Stable Diffusion (SD) and Vision-Language Models (VLMs) .

**Unlearn SD**. Diffusion models, known for generating high-quality images from text, use a text encoder (e.g., CLIP ViT-H/14 in SDv2.1) to embed prompts into a high-dimensional space. The text embedding guides the denoising process through cross-attention, starting from an initial noise $\mathbf{x}_T$ and iteratively denoising at each step:

$$\mathbf{x}_{t-1} = \sqrt{\alpha_t}\left(\mathbf{x}_t - \gamma_t \nabla_\mathbf{x} \log p(\mathbf{x}_t|\mathbf{e})\right) + \sqrt{1-\alpha_t}\mathbf{z}_t \tag{9}$$

where $\mathbf{x}_t$ is the noisy image at step $t$, $\mathbf{z}_t$ is the noise added at step $t$, $\alpha_t$ is a time-dependent parameter controlling the noise balance, $\gamma_t$ is the learning rate, $\mathbf{e} = f_{\mathtt{t}}(\mathtt{text})$ is the text embedding, and $\nabla_\mathbf{x} \log p(\mathbf{x}_t|\mathbf{e})$ is the gradient of the log-probability of the noisy image given the text embedding, guiding the denoising process. We freeze the CLIP vision model and only update the language model to achieve unlearning.

**Unlearn VLMs**. VLMs enable LLMs to process multi-modal information. LLaVA-1.5 [24] uses a pretrained CLIP vision encoder ViT-L/14-336px to extract the visual features $\mathbf{e} = f_\mathbf{v}(\mathtt{img})$, which are projected as visual tokens $\mathbf{H_v} = \mathbf{W} \cdot \mathbf{e}$ through an MLP $\mathbf{W}$. These tokens are then concatenated with language tokens $\mathbf{H_q}$ as input $\mathbf{H} = [\mathbf{H_v}; \mathbf{H_q}]$ to the language model. Since VLMs rely on the vision encoder, unlearning specific concepts in the CLIP vision model can directly influence the language model's output.

## 4 Experiment

### 4.1 Experiment setup

**Models.** We mainly experiments on CLIP and CLIP-based generative model SD, VLMs. For CLIP, we used architecture `ViT-B-32` trained on LAION-400M dataset [33], and pre-trained weights from the OpenCLIP [6]. For SD, we used the SDv2.1 from StabilityAI that built on the `CLIP-ViT-H-14` text encoder, pre-trained on the LAION-5B dataset.

**Datasets.** We used publicly-available datasets to construct the forget, retain, and evaluation sets. For unlearning target identities, we curated the forget set by filtering the LAION-400M dataset to isolate 1,000 to 6,000 image-text pairs per identity. The retain set consists of a single shard from LAION-400M, containing approximately 7,900 images (due to expiring URLs). To assess unlearning effectiveness, we used the CelebA dataset [26], sampling 100 frequently appearing celebrities from LAION-400M. Post-unlearning, model utility was evaluated using the ImageNet dataset for zero-shot classification.

**Evaluation metrics.** For CLIP, we measure unlearn performance using forget accuracy, defined as the zero-shot classification accuracy on unlearned content. Following the standard zero-shot paradigm [31], predictions are based on the highest cosine similarity between image and text embeddings. The model utility retention is assessed via zero-shot accuracy on ImageNet and CelebA.

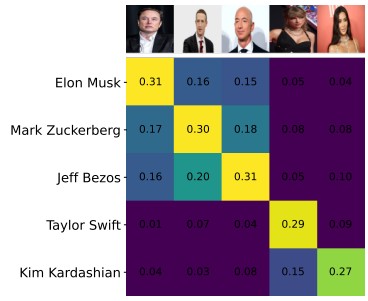 

**(a)** Original cosine similarity matrix          **(b)** Cosine similarity matrix after unlearning

**Figure 3:** Cosine similarity matrix of image-text pairs before & after unlearning "Elon Musk" as an example. (a) original CLIP correctly associate images and text of distinct identities with high similarity. (b) after unlearning, the image-text pair of "Elon Musk" is no longer matched, while other identities are only slightly affected.

**Table 1:** Performance comparison of different unlearning methods on CLIP zero-shot classification. FA@{1, 5} stands for top-{1, 5} forget accuracy (%), i.e., accuracy of unlearned identity. TA_IN@1 and TA_CA@1 stands for the top-1 test accuracy (%) on ImageNet and CelebA dataset, respectively. $K$ and $k$ denotes the number of epochs for training and iterations for unlearning, respectively ($K = 32$ and $k = 10$ in our experiments). $N$ is the size of whole training set, which is much larger than our sampled forget set ($N_f$) and retain set ($N_r$).

| Method | FA@1 ($\downarrow$) | FA@5 ($\downarrow$) | TA_IN@1 ($\uparrow$) | TA_CA@1 ($\uparrow$) | Compute Time ($\mathcal{O}$) |
|---|---|---|---|---|---|
| Original | 73.05 | 92.22 | 60.12 | 61.38 | $\mathcal{O}(K \cdot N)$ |
| learning rate = $10^{-6}$ / $10^{-7}$ | | | | | |
| FT [36] | 66.08/70.50 | 90.10/92.22 | 60.36/60.26 | 60.70/61.35 | $\mathcal{O}(k \cdot N_r)$ |
| GA [35] | 0.00/0.00 | 0.00/4.91 | 35.88/60.03 | 24.92/53.86 | $\mathcal{O}(k \cdot N_f)$ |
| GAFT (equation 1) | 0.00/2.67 | 0.00/15.89 | 55.52/60.13 | 25.71/55.22 | $\mathcal{O}(k \cdot (N_f + N_r))$ |
| SalUn [8] | 0.00/3.33 | 0.00/15.69 | 55.45/60.26 | 26.11/55.81 | $\mathcal{O}(N_f) + \mathcal{O}(k \cdot (N_f + N_r))$ |
| SSD [9] | 0.00 | 0.00 | 51.84 | 35.96 | $\mathcal{O}(N_f + N_r)$ |
| SLUG (ours) | 0.00 | 0.00 | 59.96 | 58.32 | $\mathcal{O}(N_f + N_r)$ |

**Comparing methods.** We compare with the state-of-the-art methods along with classical methods. For unlearning in CLIP models, we compare with classical fine tuning (FT) [36], gradient ascent (GA) / negative gradient (NG) [35], and recent salient parameters based saliency unlearning (SalUn) [8], and selective synaptic dampening (SSD) [9].

### 4.2 Unlearning for CLIP

**Unlearning identities.** We demonstrate that modifying a single layer suffices to unlearn an identity or concept while preserving the model's overall utility. Figure 3 presents an example of unlearning targeted identity "Elon Musk" on CLIP. Each cell in matrices shows the cosine similarity between the embeddings of an image-text pair. Before unlearning (Figure 3a), high similarity values are obsered along the diagonal, indicating strong alignment between images and corresponding text descriptions across all identities. After unlearning (Figure 3b), the similarity of targeted identity image-text pairs decrease, while other identities remain largely unaffected. This demonstrates SLUG's precision in selectively removing specific information. Additional studies on other identities and model architectures are presented in the Appendix Figure 6 and 8. Moreover, Figure 8 showcases SLUG's capability to simultaneously unlearn multiple identities, highlighting its scalability and flexibility.

**Unlearning without losing utility.** A key advantage of SLUG is that performance on unrelated tasks remains intact. Table 1 presents quantitative performance comparisons of various methods for classification on ImageNet and CelebA. For CelebA, unlearning an identity slightly reduces accuracy due to its close relationship with the data distribution. As shown in Table 1, our method outperforms others in forget and retain accuracy while maintaining minimal computational complexity, requiring only a one-time gradient

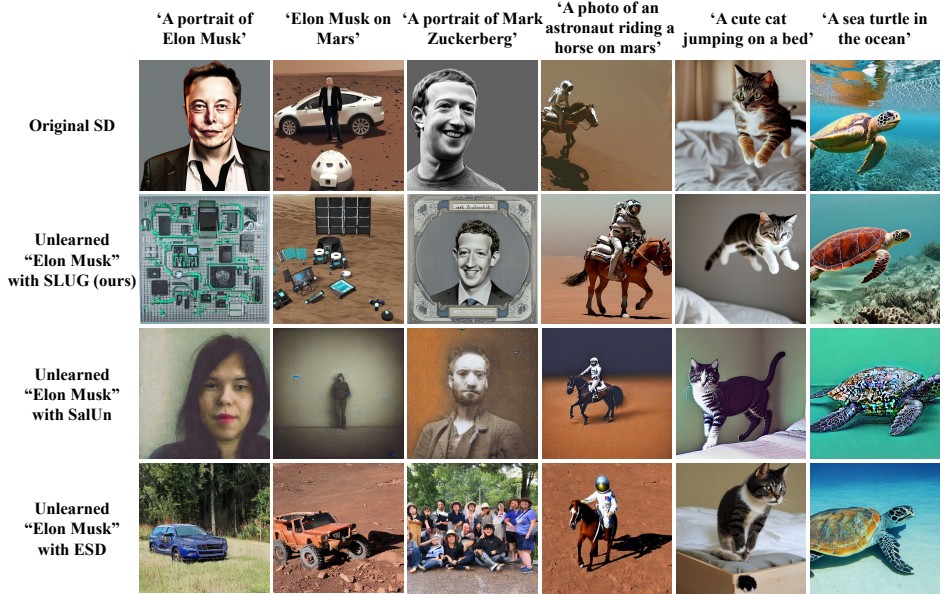

**Figure 4:** Images generated by different SDs using column captions as prompts. First row: images generated by the original pretrained SD. Second row: outputs of the SD after "Elon Musk" is unlearned using SLUG. We can see that "Elon Musk" is effectively unlearned, whereas other objects are unaffected. Bottom two rows: outputs of the SDs after "Elon Musk" is unlearned by existing methods (SalUn and ESD). We observe images generated for other unrelated prompts are also affected to some degree.

computation ($\mathcal{O}(N_{\mathtt{f}} + N_{\mathtt{r}})$) for unlearning. In contrast, other methods need $k$ iterative gradient calculations and careful hyperparameter tuning, such as learning rate, to balance unlearning effectiveness and utility preservation. For instance, a high learning rate (e.g., $10^{-6}$) compromises utility, while a low rate (e.g., $10^{-7}$) reduces unlearning effectiveness.

**Localizing layers.** Our method efficiently identifies critical layers for unlearning, reducing the search space from hundreds to just a few. Figures 2, 7, and 12 show which layers are selected for unlearning different identities. This is achieved by combining two key metrics: layer importance, which measures how sensitive the forget loss is to changes in each layer, and gradient alignment, ensuring updates minimally affect the retain set. These metrics help identify Pareto-optimal layers that balance effective unlearning with preserving model utility (explained further in Section 3). The late attention layers in vision models and early attention layers in language models are targeted for updates because they play critical roles in refining high-level features and establishing foundational understanding, respectively. In vision transformers, late layers focus on contextually rich features, while in language models, early layers process key sequential and contextual dependencies.

## 4.3 Unlearning for Stable Diffusion

**Unlearning identity.** Stable Diffusion (SD) models excel in text comprehension and image generation, producing high-fidelity results such as "a portrait of Elon Musk." Adjusting the prompt can create imaginative content, such as "Elon Musk on Mars." However, their potential misuse raises concerns about harm to data providers [40]. This study demonstrates how to erase personal information from an image generation model, ensuring prompts for the erased individual produce inaccurate results. Figure 4 shows examples before and after unlearning. Our method, when applied to Elon Musk, generates electronic circuits consistently, without reducing the model's ability to produce diverse objects. In contrast, other methods degrade both the portraits of others and the quality of unrelated images. We provide additional results on unlearning more celebrity identities, and other case studies on unlearning copyright-protected content and novel concept, in Section G.

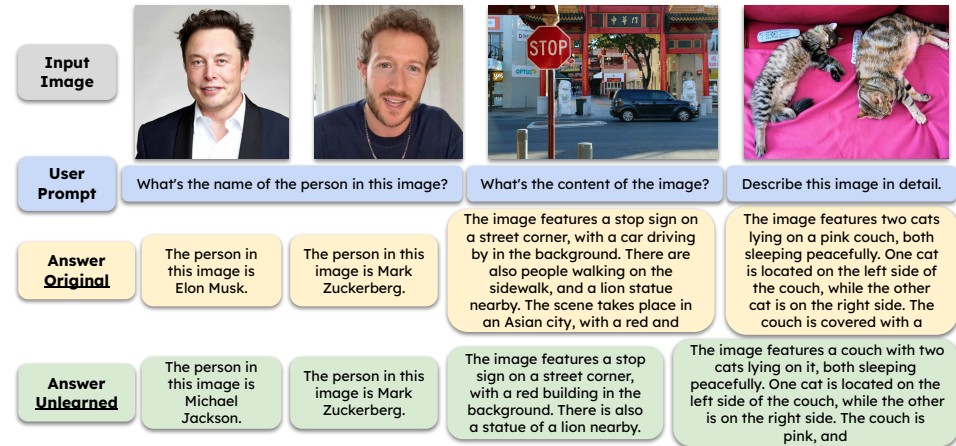

**Figure 5:** Effects of SLUG unlearning "Elon Musk" on LLaVA 1.5. The third row with yellow boxes shows the answers of the original model, and the forth row with green boxes shows the output of the unlearned model, where Elon Musk is effectively unlearned, whereas other concepts are unaffected.

## 4.4 Unlearning for VLMs

VLMs excel in visual understanding and question answering, accurately identifying individuals in images. Figure 5 shows that when given an image of Elon Musk and asked, "What's the name of the person in this image?", the model correctly names him.

Our experiments focused on the VLM model LLaVA-1.5, which uses a pre-trained CLIP visual encoder to extract visual features from images. These features are transformed into a format compatible with the language model using a neural network layer that projects them into the word embeddings space. The resulting visual tokens are combined with language tokens and fed into the model to generate responses. The key insight of our method is that the vision capability of VLMs heavily relies on the visual encoder. Therefore, by unlearning certain concepts from the CLIP vision model, we can influence the language model's understanding and generation of responses without directly modifying the language model itself. Figure 5 demonstrates the effectiveness of our approach. When given an image of Elon Musk and asked to identify the person, the original model correctly names him. After applying our unlearning method, the model incorrectly identifies Elon Musk as Michael Jackson, indicating that the specific identity information has been successfully removed. This alteration does not significantly impact the model's overall utility. Additional examples of this phenomenon are discussed in Section H.

## 5 Conclusion

In this work, we introduced SLUG, an efficient machine unlearning method that requires just a single gradient computation and updates only one layer of the model. SLUG enhances unlearning feasibility on large models, especially with constrained hardware, while preserving overall model utility. Our experiments with CLIP, and Stable Diffusion show that SLUG outperforms existing methods, particularly in unrelated tasks, with minimal computational overhead. The key innovation of SLUG is its ability to identify and update only the most relevant layers for the desired unlearning concepts, which also provides new insights into the internal representations learned by different parts of neural networks. This contributes to the ongoing effort to improve the interpretability and transparency of AI systems.

## Acknowledgments

This work used Jetstream2 [13] at Indiana University through allocation CIS220128 from the Advanced Cyberinfrastructure Coordination Ecosystem: Services & Support (ACCESS) program [4], which is supported by NSF grants #2138259, #2138286, #2138307, #2137603, and #2138296.

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

## Appendix A  More examples on unlearning identities

In addition to the experiment on unlearning "Elon Mask" identity in the CLIP model, as discussed in Sec. 4.2 of the main text, we performed similar experiment on a broader set of identities: {Kanye West, Barack Obama, Bruce Lee, Fan Bingbing, Lady Gaga}. These names were selected from the CelebA dataset to represent a diverse cross-section of ethnicities and genders. Our method effectively identified the crucial layers associated with each name. These layers can then be specifically targeted to efficiently unlearn the corresponding identity from the CLIP model.

Figure 6 demonstrates that our approach successfully removes the desired names from the CLIP model (i.e., image-text alignment or cosine similarity becomes extremely low) . Figure 7 illustrates the Pareto-front plots that are used to identify important layers selected by our method for unlearning different identities.

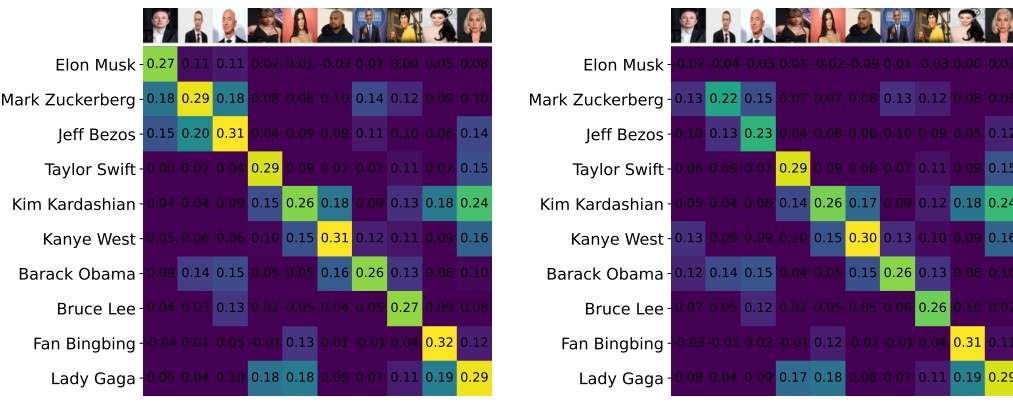

(a) Original cosine similarity matrix    (b) Cosine similarity matrix after unlearning

**Figure 6:** Cosine similarity matrix of image and text pairs before and after unlearning Elon Musk. After unlearning, the image and text pair of Elon Musk are not matched, while other persons are only slightly affected. Here the `vision attention out projection layer at the` $9_{th}$ `resblock` (associate with `9.attn.out_proj` in the pareto front legend) is unlearned. CLIP model: `ViT-B-16`

## Appendix B  Joint update for unlearning multiple identities

We study the composite effect of our approach where we unlearn multiple tasks simultaneously. For instance, in the task of unlearning multiple identities, we use the gradients calculated for each identity on the original model and corresponding forget sets to identify the layers that are most significant for the respective identities, and then perform layer updates to simultaneously unlearn all of them. For joint updating, we follow the same updating scheme as described in Sec. 3. Firstly, different identities have different step size initialization from their respective gradients, and later on the step size is updated separately using binary search based on the unlearning result of the respective identity. We present our results in Figure. 8, where we successfully unlearn (a) {Elon Musk, Mark Zuckerberg} and (b) {Elon Musk, Taylor Swift}.

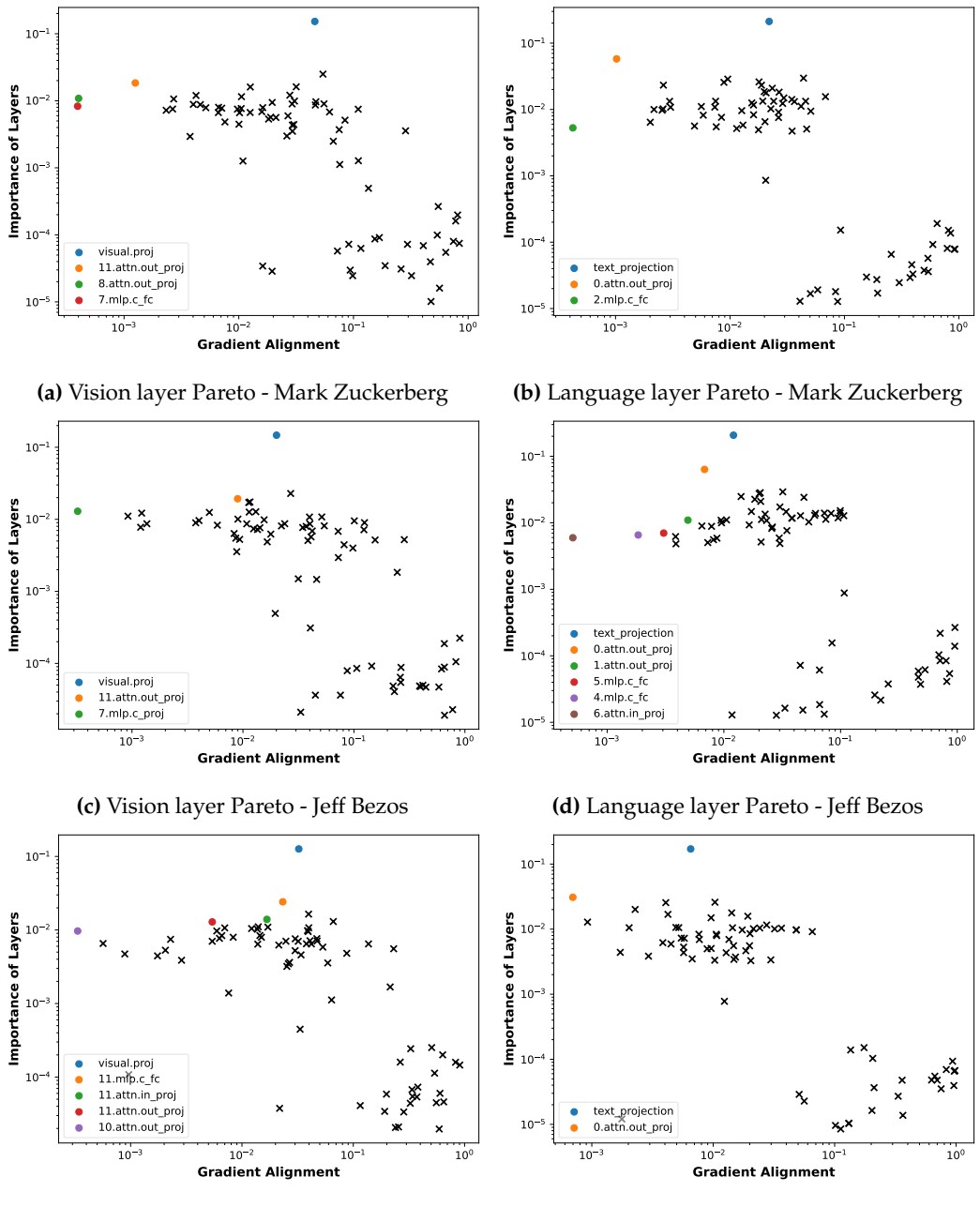

**(a)** Vision layer Pareto - Mark Zuckerberg

**(b)** Language layer Pareto - Mark Zuckerberg

**(c)** Vision layer Pareto - Jeff Bezos

**(d)** Language layer Pareto - Jeff Bezos

**(e)** Vision layer Pareto - Taylor Swift

**(f)** Language layer Pareto - Taylor Swift

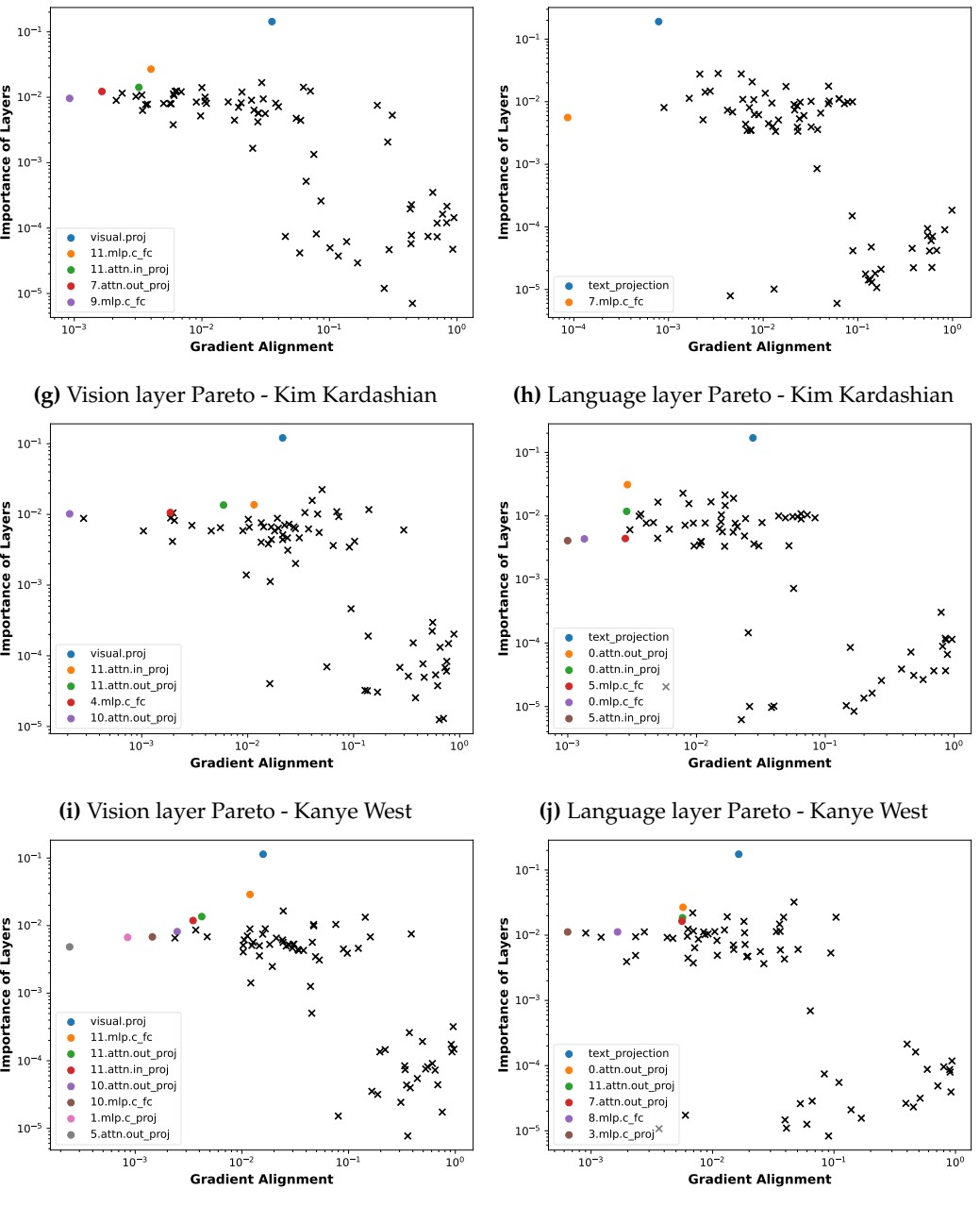

**(g)** Vision layer Pareto - Kim Kardashian

**(h)** Language layer Pareto - Kim Kardashian

**(i)** Vision layer Pareto - Kanye West

**(j)** Language layer Pareto - Kanye West

**(k)** Vision layer Pareto - Barack Obama

**(l)** Language layer Pareto - Barack Obama

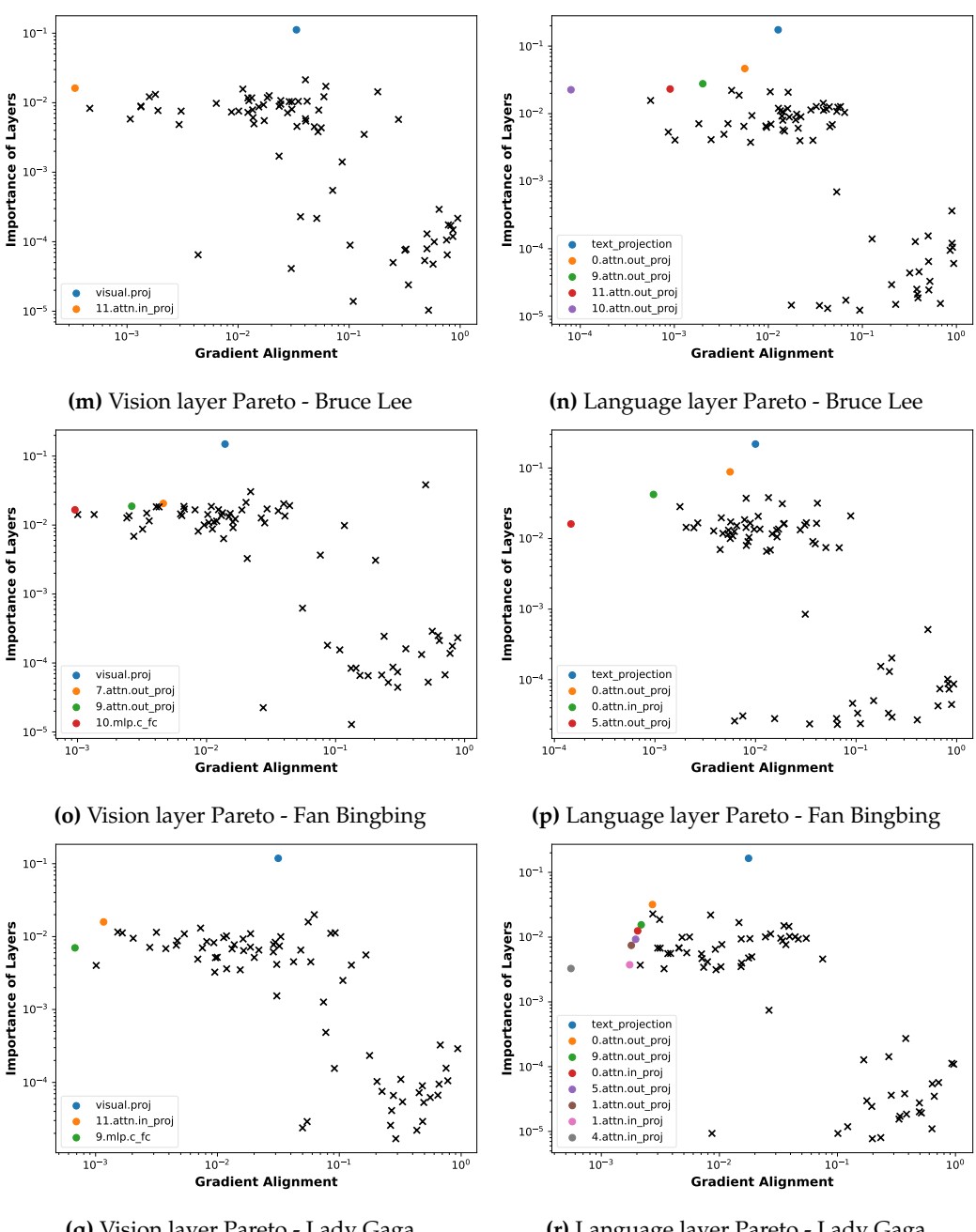

**(m)** Vision layer Pareto - Bruce Lee

**(n)** Language layer Pareto - Bruce Lee

**(o)** Vision layer Pareto - Fan Bingbing

**(p)** Language layer Pareto - Fan Bingbing

**(q)** Vision layer Pareto - Lady Gaga

**(r)** Language layer Pareto - Lady Gaga

**Figure 7:** Scatter plots of layers for unlearning more identities, same setting as Figure 2. CLIP model `ViT-B-32`. Figures (a) - (r) shows the importance and gradient alignment of different vision model and language model layers as we unlearn different identities.

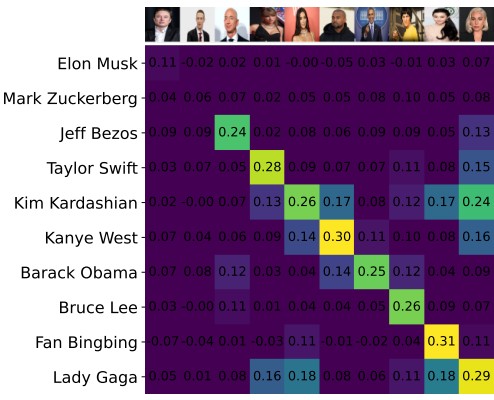
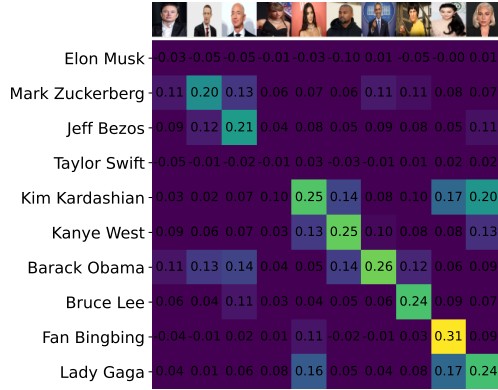

**(a)** Cosine similarity matrix after unlearning Elon Musk and Mark Zuckerberg

**(b)** Cosine similarity matrix after unlearning Elon Musk and Taylor Swift

**Figure 8:** Cosine similarity matrix of image and text pairs after unlearning multiple name identities (see Figure. 6a for cosine similarity matrix on original model). (a) Unlearning Elon Musk and Mark Zuckerberg. (b) Unlearning Elon Musk and Tylor Swift. In both cases, the image and text pair of selected identities are not matched after unlearning, while other identifies are only slightly affected. We selected and updated the vision layer `9.attn.out_proj` for Elon Musk and the vision layer `11.attn.out_proj` for the other identity according to the pareto fronts in Fig. 7a and Fig. 7e, in both (a) and (b). We used CLIP model: `ViT-B-32` for these experiments.

We also investigate how the unlearning performance varies as the number of identities to be forgotten increases. The identified layers are then updated in parallel to achieve unlearning of $N$ identities. Figure 9 demonstrate the effectiveness of our approach in unlearning $N$ identities for different values of $N$. Figure 7 presents details on identifying layers associated with different identities and updating them to achieve unlearning of multiple identities at once.

## Appendix C   More CLIP models

We performed experiments using an expanded set of model architectures. The results for {`ViT-B-16` are discussed above in Figure 6. The results for `ViT-L-14, EVA01-g-14`} are discussed in Figures 10,11, respectively. Figure 12 shows the metrics for different layers that our method uses to identify significant layers. These results demonstrate our method offers scalability and effectiveness across a range of model sizes, from 149.62 million parameters (`ViT-B-16`) to 1.136 billion parameters (`EVA01-g-14`). This underscores the flexibility of our approach to accommodate models of different scales.

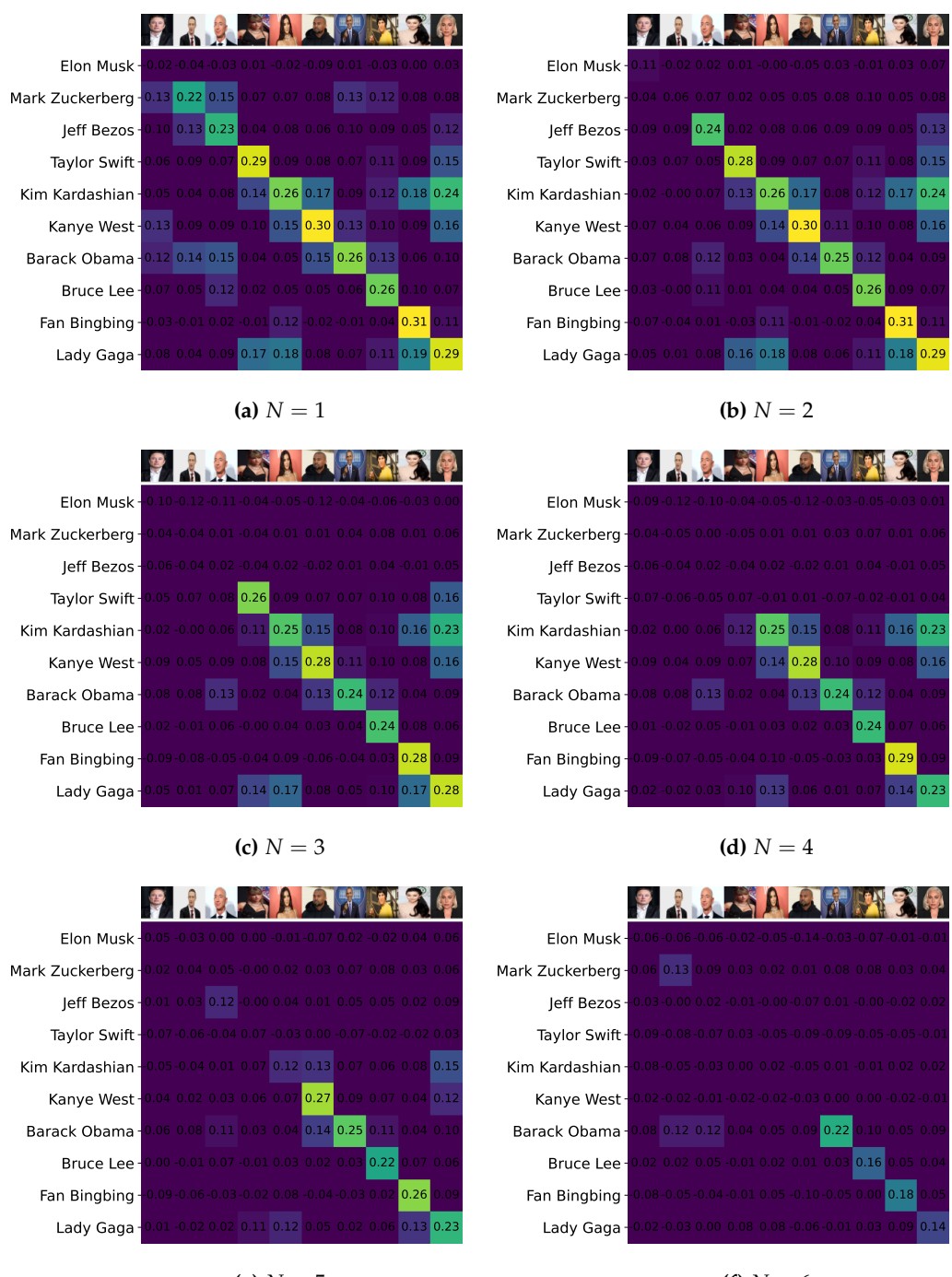

**(a)** $N = 1$        **(b)** $N = 2$

**(c)** $N = 3$        **(d)** $N = 4$

**(e)** $N = 5$        **(f)** $N = 6$

**Figure 9:** Cosine similarity matrices as we unlearn $N$ identities, where $N \in \{1, 2, ..., 6\}$. (a)–(f) Unlearn Elon Musk, Mark Zuckerberg, Jeff Bezos, Taylor Swift, Kim Kardashian, and Kanye West in a joint manner. To unlearn $N$ identities, our method (SLUG) identifies up to $N$ layers in the model using the single gradient calculated with the original network weights. The identified layers are then updated in parallel to achieve unlearning of $N$ identities.

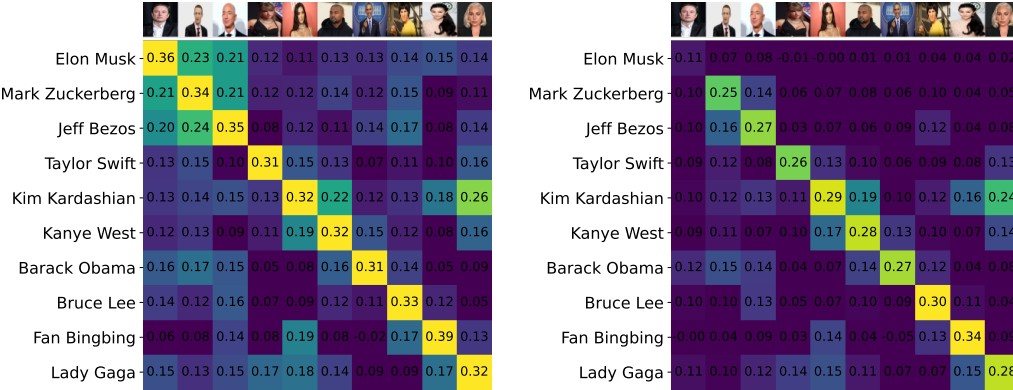

**(a)** Original cosine similarity matrix

**(b)** Cosine similarity matrix after unlearning

**Figure 10:** Cosine similarity matrix of image and text pairs before and after unlearning Elon Musk. After unlearning, the image and text pair of Elon Musk are not matched, while other persons are only slightly affected. Here, based on the pareto front in Fig. 12c, we select and update the vision layer `23.mlp.c_fc` for unlearning. CLIP model: `ViT-L-14`

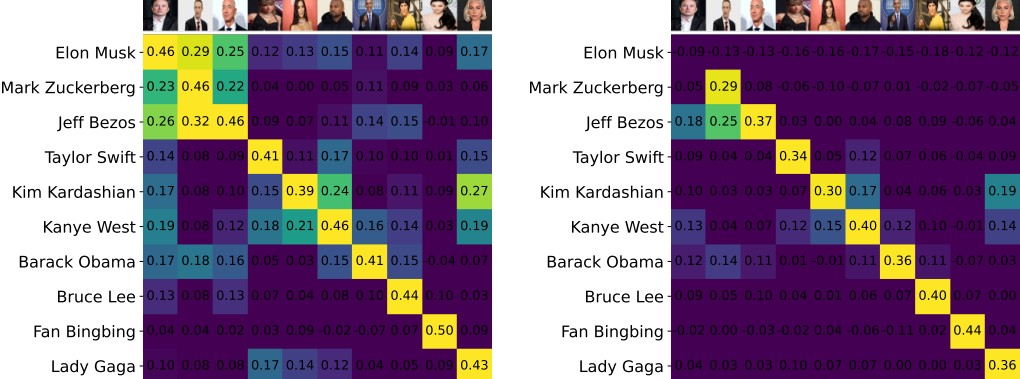

**(a)** Original cosine similarity matrix

**(b)** Cosine similarity matrix after unlearning

**Figure 11:** Cosine similarity matrix of image and text pairs before and after unlearning Elon Musk. After unlearning, the image and text pair of Elon Musk are not matched, while other persons are only affected. Here, based on the pareto front in Fig. 12f, we select and update the language layer `11.attn.out_proj` for unlearning. CLIP model: `EVA01-g-14`.

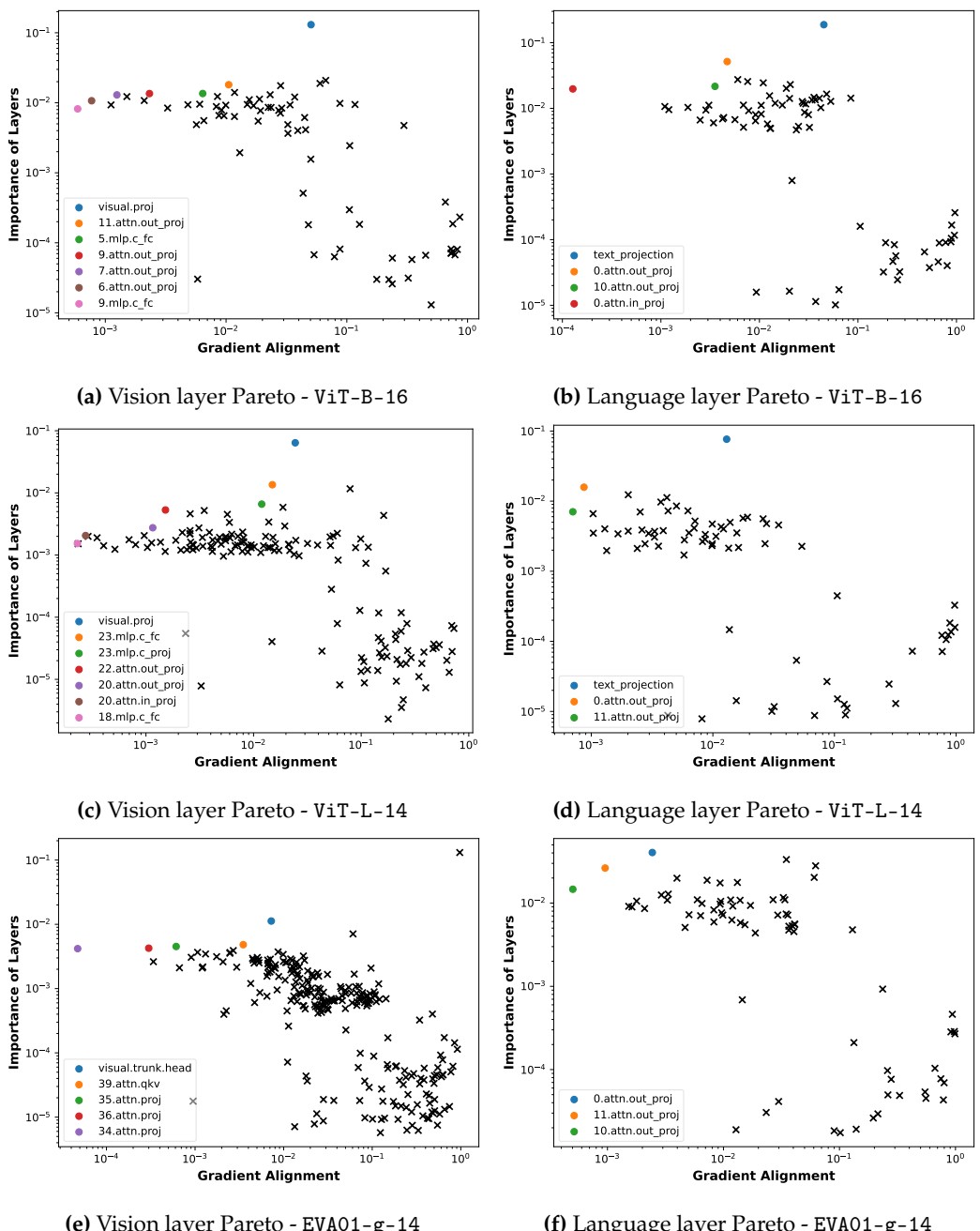

**(a)** Vision layer Pareto - `ViT-B-16`

**(b)** Language layer Pareto - `ViT-B-16`

**(c)** Vision layer Pareto - `ViT-L-14`

**(d)** Language layer Pareto - `ViT-L-14`

**(e)** Vision layer Pareto - `EVA01-g-14`

**(f)** Language layer Pareto - `EVA01-g-14`

**Figure 12:** More CLIP models, in addition to Sec 4.2. Unlearning name `Elon Musk` from different CLIP models built in: {`ViT-B-16, ViT-L-14, and EVA01-g-14`}

# Appendix D Unlearn different concepts

In addition to unlearning identities from CLIP, we also sample 7 classes {Basketball, Beach, Castle, Revolver, Rifle, School bus, Sunglasses} from ImageNet to evaluate the unlearning performance of our method on object concepts. For this experiment, we use 10k ImageNet validation images and sample images associated with target classes to create forget sets and compute gradients to unlearning different classes from the CLIP model. For evaluation, we use zero-shot accuracy reduction as the metric of effective unlearning target classes from the CLIP. The results, presented in Table. 2, show the CLIP zero-shot accuracy evaluations

for both the forgetting of sampled classes and the retention of other ImageNet classes after unlearning. Our findings indicate that our method effectively reduces the CLIP zero-shot accuracy for the targeted classes to 0.0%, while the accuracy for remaining classes remains high, experiencing only minimal degradation (ranging from 0.03% to 2.03%) compared to the original pre-trained model, which indicates that the model's original functions are highly preserved after our unlearning.

**Table 2:** Unlearning performance of our method on common object concepts. FA@1 and FA@5 represents the top-1 and top-5 forget accuracy (%) of each forget class (i.e., zero-shot classification accuracy of unlearned class). TA@1 and TA@5 represents the top-1 and top-5 accuracy (%) of all classes of ImageNet except the corresponding Forget class. Each row shows the forget class accuracy and average accuracy over all classes of ImageNet before and after unlearning a class. Our method can reduce the forget accuracy of Forget classes to 0.0% while keeping the accuracy of the remaining classes close to original model (within $0.06 - 2.03\%$ difference). CLIP model: `ViT-B-32`. TA@1 and TA@5 for the original model remains almost the same for all rows; therefore, we list it once in the table.

| Forget class | Original | | | | Unlearned | | | |
|---|---|---|---|---|---|---|---|---|
| | FA@1 | FA@5 | TA@1 | TA@5 | FA@1 ↓ | FA@5 ↓ | TA@1 ↑ | TA@5 ↑ |
| Basketball | 100.0 | 100.0 | | | 0.0 | 0.0 | 59.18 | 84.48 |
| Beach | 54.55 | 72.73 | | | 0.0 | 0.0 | 59.54 | 84.78 |
| Castle | 87.50 | 100.0 | | | 0.0 | 0.0 | 58.13 | 83.87 |
| Revolver | 100.0 | 100.0 | 60.16 | 85.52 | 0.0 | 0.0 | 59.94 | 85.43 |
| Rifle | 42.86 | 57.14 | | | 0.0 | 0.0 | 60.08 | 85.49 |
| School bus | 76.92 | 100.0 | | | 0.0 | 0.0 | 59.50 | 89.18 |
| Sunglasses | 44.44 | 55.56 | | | 0.0 | 0.0 | 60.13 | 85.23 |

# Appendix E  Linearity of unlearning trajectory of different layers

In addition to the layers presented in Figure 2 (c) and (d), we show in Figure 13 that different layers show similar unlearning behaviors if we update them along their respective gradient direction (computed once for the original model). Nevertheless, the utility performance may vary depending on the selected layer; thus, it is important to select the best layer from the Pareto set for the overall best performance.

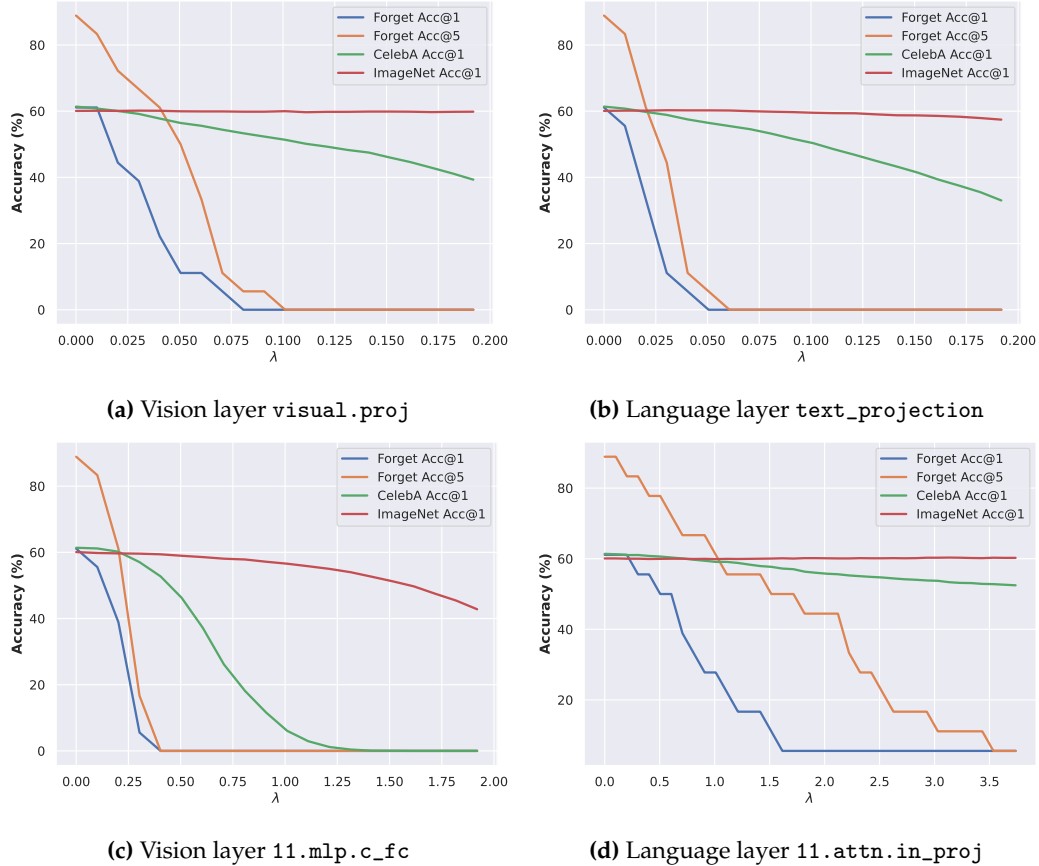

**(a)** Vision layer `visual.proj`

**(b)** Language layer `text_projection`

**(c)** Vision layer `11.mlp.c_fc`

**(d)** Language layer `11.attn.in_proj`

**Figure 13:** More examples of unlearning different layers. Correspond to Figure 2. The performance changes monotonically with the step size $\lambda$.

**Evaluation on UnlearnCanvas benchmark.** To demonstrate the unlearning effectiveness and efficiency of SLUG, we also evaluate its performance on UnlearnCanvas [44], a benchmark focused on unlearning artistic style and object concepts in Stable Diffusion. It introduces a comprehensive set of metrics for both evaluating effectiveness and efficiency, including **UA** (Unlearn Accuracy), **IRA** (In-domain Retain Accuracy), and **CRA** (Cross-domain Retain Accuracy). The benchmark targets unlearning styles and objects from an SDv1.5 model fine-tuned to generate 20 different objects in 60 distinct styles. The benchmark utilizes target SD with the prompt: "A [`object name`] in [`style name`] style," to generate the unlearning dataset, comprising 20 images for each object-style pair (i.e., 400 images per style and 1,200 images per class), resulting in 24,000 images in total. We curate forgets set with images associated with each style/object for each unlearning objective.

In Table 3, we report the unlearning performance of SLUG in benchmark metrics, along with other state-of-the-art unlearning methods reported in UnlearnCanvas. Our method minimizes storage and computational time by only requiring the gradient values of a few layers on the Pareto front to be stored, and performing a one-step update along the gradient for unlearning. Despite being extremely efficient, our method does not suffer from significant performance degradation in any metric or task in UnlearnCanvas. It achieves an optimal trade-off between unlearning and retaining accuracy compared to other unlearning methods. For qualitative evaluation, we provide visual examples in Section G.

## Appendix F    Experiment details on UnlearnCanvas

**Models.** UnlearnCanvas targets unlearning styles and objects from an SDv1.5 model fine-tuned to generate 20 different objects in 60 distinct styles. The benchmark provides

**Table 3:** Performance overview of different unlearning methods on UnlearnCanvas. The best performance for each metric is highlighted in green, and significantly underperforming results, in benchmark criteria, are marked in red. Our method SLUG shows no significant underperforming, and achieves the best trade-off among unlearning, retaining, and efficiency.

| Method | Effectiveness | | | | | | | Efficiency | | |
| --- | --- | --- | --- | --- | --- | --- | --- | --- | --- | --- |
| | Style Unlearning | | | Object Unlearning | | | FID (↓) | Time | Memory | Storage |
| | UA (↑) | IRA (↑) | CRA (↑) | UA (↑) | IRA (↑) | CRA (↑) | | (s) (↓) | (GB) (↓) | (GB) (↓) |
| ESD [10] | 98.58% | 80.97% | 93.96% | 92.15% | 55.78% | 44.23% | 65.55 | 6163 | 17.8 | 4.3 |
| FMN [42] | 88.48% | 56.77% | 46.60% | 45.64% | 90.63% | 73.46% | 131.37 | 350 | 17.9 | 4.2 |
| UCE [11] | 98.40% | 60.22% | 47.71% | 94.31% | 39.35% | 34.67% | 182.01 | 434 | 5.1 | 1.7 |
| CA [20] | 60.82% | 96.01% | 92.70% | 46.67% | 90.11% | 81.97% | 54.21 | 734 | 10.1 | 4.2 |
| SalUn [8] | 86.26% | 90.39% | 95.08% | 86.91% | 96.35% | 99.59% | 61.05 | 667 | 30.8 | 4.0 |
| SEOT [23] | 56.90% | 94.68% | 84.31% | 23.25% | 95.57% | 82.71% | 62.38 | 95 | 7.34 | 0.0 |
| SPM [27] | 60.94% | 92.39% | 84.33% | 71.25% | 90.79% | 81.65% | 59.79 | 29700 | 6.9 | 0.0 |
| EDiff [38] | 92.42% | 73.91% | 98.93% | 86.67% | 94.03% | 48.48% | 81.42 | 1567 | 27.8 | 4.0 |
| SHS [37] | 95.84% | 80.42% | 43.27% | 80.73% | 81.15% | 67.99% | 119.34 | 1223 | 31.2 | 4.0 |
| SLUG (Ours) | 86.29% | 84.59% | 88.43% | 75.43% | 77.50% | 81.18% | 75.97 | 39 | 3.61 | 0.04 |

pre-trained SDv1.5 models for evaluation in `Diffusers` and `CompVis` implementations. In our experiment, correspondly, we focus on the CLIP text encoder used in SDv1.5 `Diffusers` implementation: `openai/clip-vit-large-patch14` from HuggingFace.

**Computational time, memory, and storage.** The gradient computational time and memory usage of SLUG depends on several factors: computing resource, batch size, and size of the forget set. Note that while the details of the evaluation of efficiency metrics are not well defined in the original UnlearnCanvas, in Table. 3, we are reporting the best performance of SLUG can achieve on our computing resource `NVIDIA A100`. Specifically, the batch size is set to 1 in order to reproduce the memory usage of SLUG, and set to 16 in order to reproduce the computational time of SLUG, which is the same setting as other experiments.

For SLUG storage consumption, as our method only requires storing the gradient values of a few layers on the Pareto front, the actual storage consumption is 43 MB (0.043 GB), which by approximation is 0.0 GB in the original benchmark scale.

# Appendix G  More examples on Stable Diffusion

To demonstrate the performance and practical utility of our method, we further consider unlearning more celebrity names and more scenarios including unlearning copyright characters, novel concepts and artistic styles on Stable Diffusion.

**More celebrity names.** Beyond unlearning "Elon Musk" from Stable Diffusion, which is presented in the Figure 4, here we also provide additional qualitative evaluations on unlearning other celebrity names {Taylor Swift, Jeff Bezos} with our method in Figure 14.

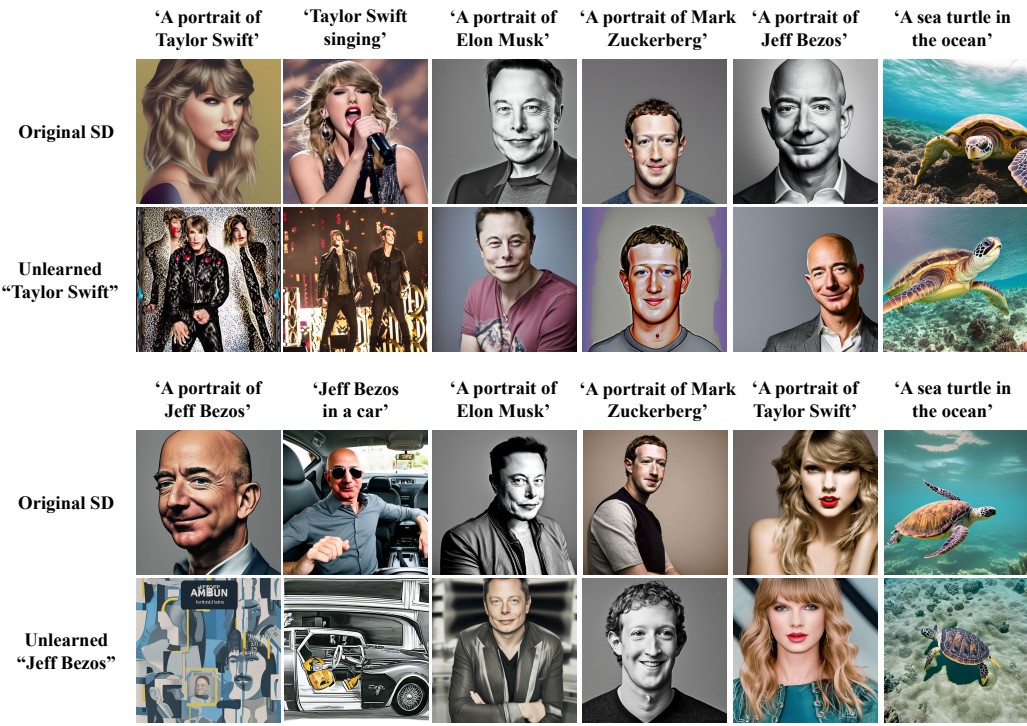

**Figure 14:** Qualitative evaluation on unlearning celebrity names Taylor Swift and Jeff Bezos from the Stable Diffusion.

**Unlearning concepts and copyright content.** In addition to identity removal for privacy protection, we address copyright concerns that increasingly challenge generative models. For unlearning copyrighted contents from Stable Diffusion models, we generate 500 images using unlearning targets as prompts, and use them as the forget set. The retain set is a single shard of LAION-400M dataset, same as for CLIP unlearning.

We successfully apply our method to remove copyright-protected content, specifically targeting well-known characters such as Marvel's "Iron Man" and Walt Disney's "Mickey Mouse." Figure 15 illustrates that our technique precisely unlearns the targeted concepts, effectively disabling the generation of images associated with these copyrighted entities while preserving the ability of the model to produce images of other concepts. These results demonstrate the use of SLUG in protecting intellectual property from generative AI.

**Novel concept.** One of the intriguing properties of the Stable Diffusion is its ability to generalize image generation to novel concepts that are infrequently or never observed in the real world. In this experiment, we explore the unlearning of a unique concept, "Avocado chair" from Stable Diffusion. We first generate 500 image using SD with the prompt "An avocado chair" to create the forget set, and use the same retain set as other experiments, which is is a single shard of LAION-400M dataset. In Figure 16, we show that our method successfully unlearn the concept "Avocado chair" from SD, resulting in the model's inability to generate images corresponding to this specific concept.

It is noteworthy that the model's capability to generate images related to the constituent atomic concepts (namely "Avocado" and "Chair") is also compromised. We hypothesize that this occurs due to the model's treatment of novel concepts as compositions of atomic concepts. For example, the concept "Avocado chair" is interpreted by the model as "Avocado" plus "Chair." Consequently, when a novel concept is unlearned, the associated atomic concepts are inadvertently affected as well. This highlights a challenge in the model's approach to handling the interoperability of novel and atomic concepts.

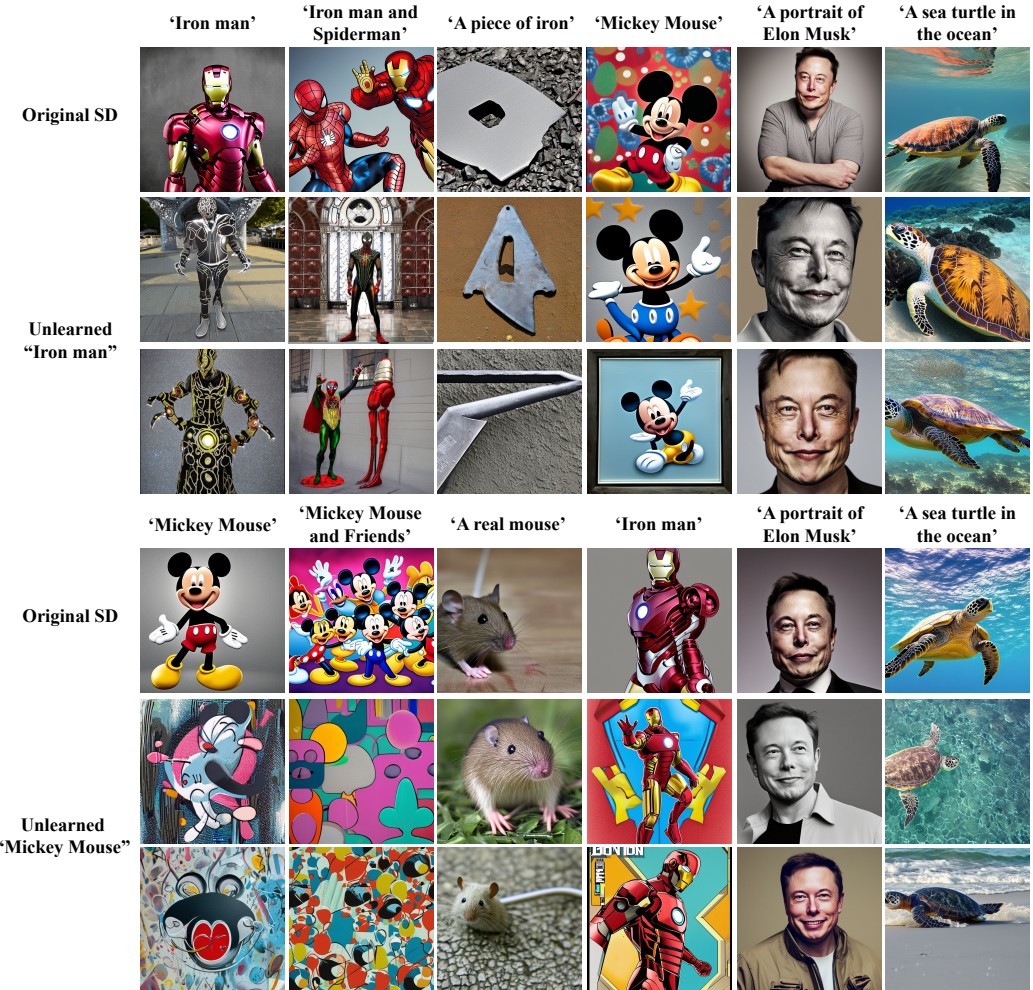

**Figure 15:** Qualitative evaluation on unlearning copyright characters "Iron man" and "Mickey Mouse" from SD, in first and second groups of figures respectively. First row shows the generated images from the original pretrained model, the second and third rows show the output of unlearned model using prompts captioned at the top of each column. Our method precisely unlearned copyright protected concepts from SD, while the image generation quality on other concepts is highly preserved.

**Artistic styles and object.** In the experiment of evaluating SLUG performance on Unlearn-Canvas benchmark discussed in Section. 4.3, we use 400 images that are associated with each style, as the forget set for unlearning style, and 1200 images that are associated with each object concept as the forget set for unlearning object, all images are from the benchmark dataset. We use a single shard of LAION-400M dataset as the retain set.

For qualitative evaluation of this experiment, we provide visual examples of unlearning artistic styles: {Pop Art, Crayon, Sketch, Van Gogh} and object: dog that are sampled from UnlearnCanvas, in Figure 18, 19 and 20. These results further show the effectiveness of SLUG in unlearning a broad spectrum of concepts ranging from concrete (e.g., celebrity name, intellectual property figure, and object) to abstract (e.g., novel concept and artistic style).

## Appendix H    More examples on VLM

In addition to results presented in the main text Figure 5, we also present additional results on unlearning a different name "Taylor Swift" from VLM in Figure 17. We demonstrate that our method can anonymize celebrity names from the pretrained Vision-language models,

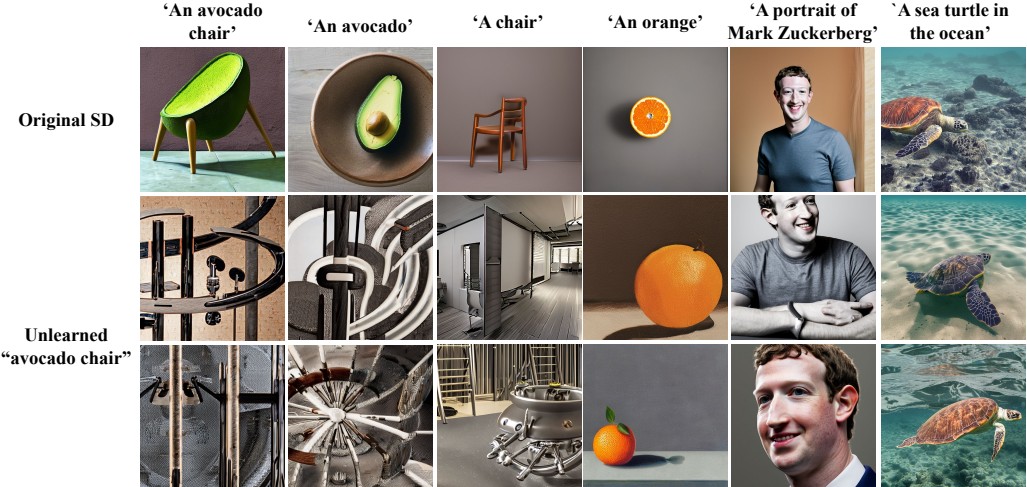

**Figure 16:** Qualitative evaluation on unlearning a novel concept "Avocado chair" from the SD.

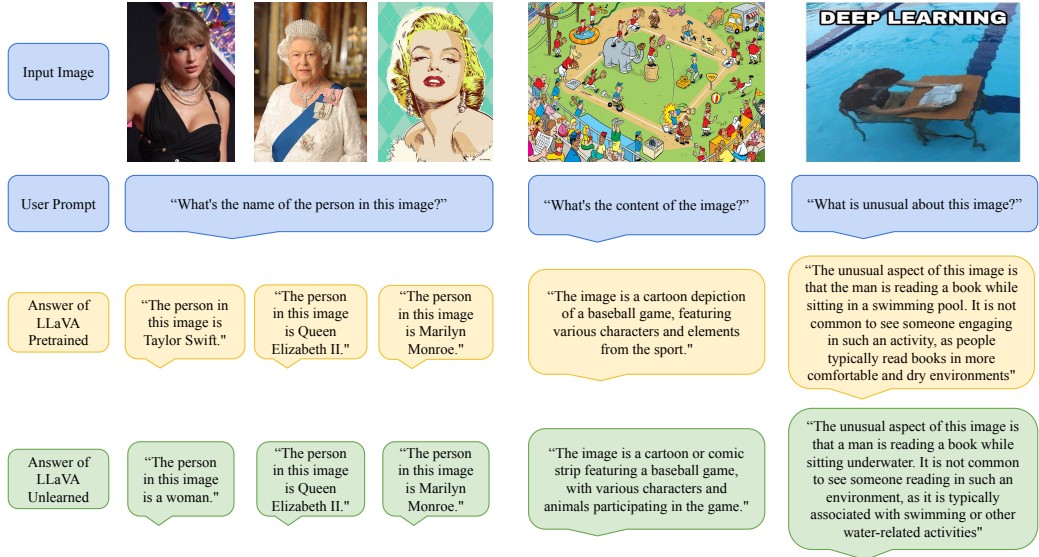

**Figure 17:** Qualitative evaluation on unlearning name "Taylor Swift" from LLaVA 1.5. While "Taylor Swift" is mapped to "woman" after the unlearning, the other female celebrity identification remain unaffected. Besides, model's robustness against style distribution shift is also preserved.

and simultaneously preserve the model's ability on image understanding, reasoning and distribution shift robustness on art work, cartoon style images.

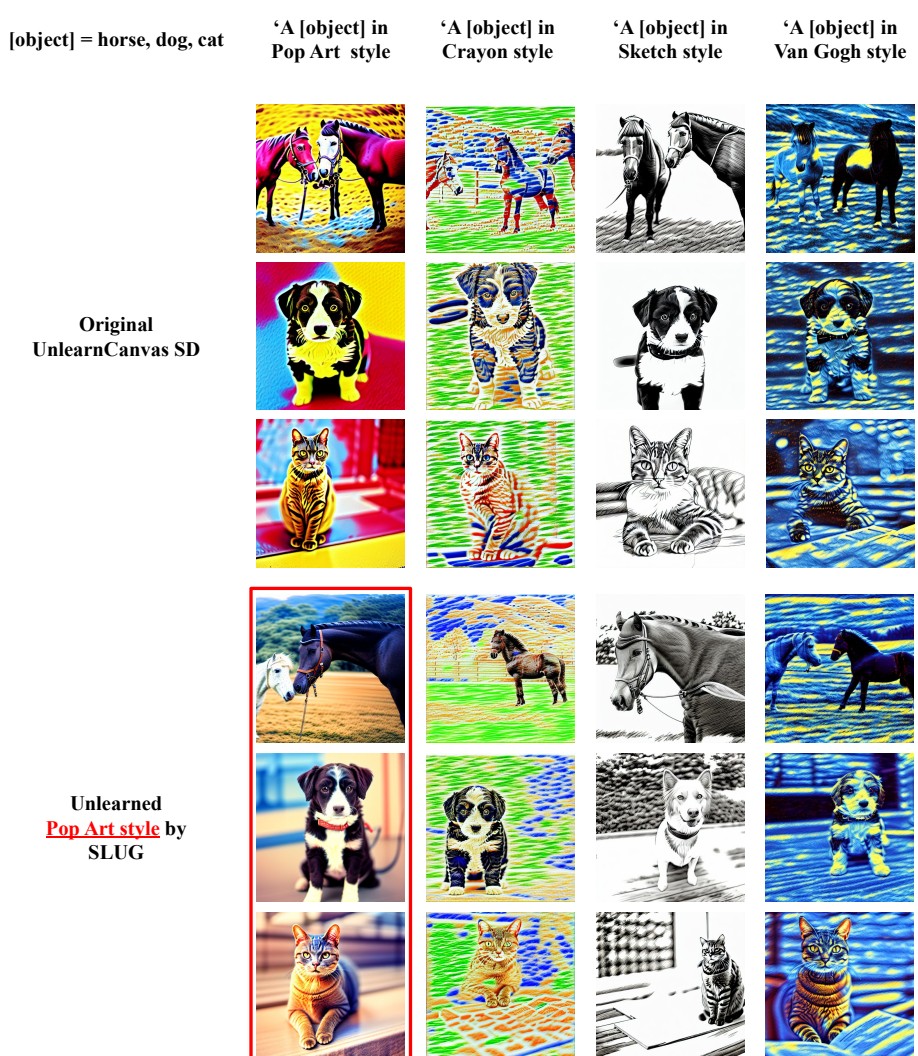

**Figure 18:** Visual examples of SLUG performance on UnlearnCanvas. Row $1-3$: outputs from original UnlearnCanvas Stable Diffusion (SD) using column captions as prompts. Row $4-6$: outputs from UnlearnCanvas SD unlearned Pop Art style. Outputs corresponding to the unlearned style are highlighted by the red bounding box.

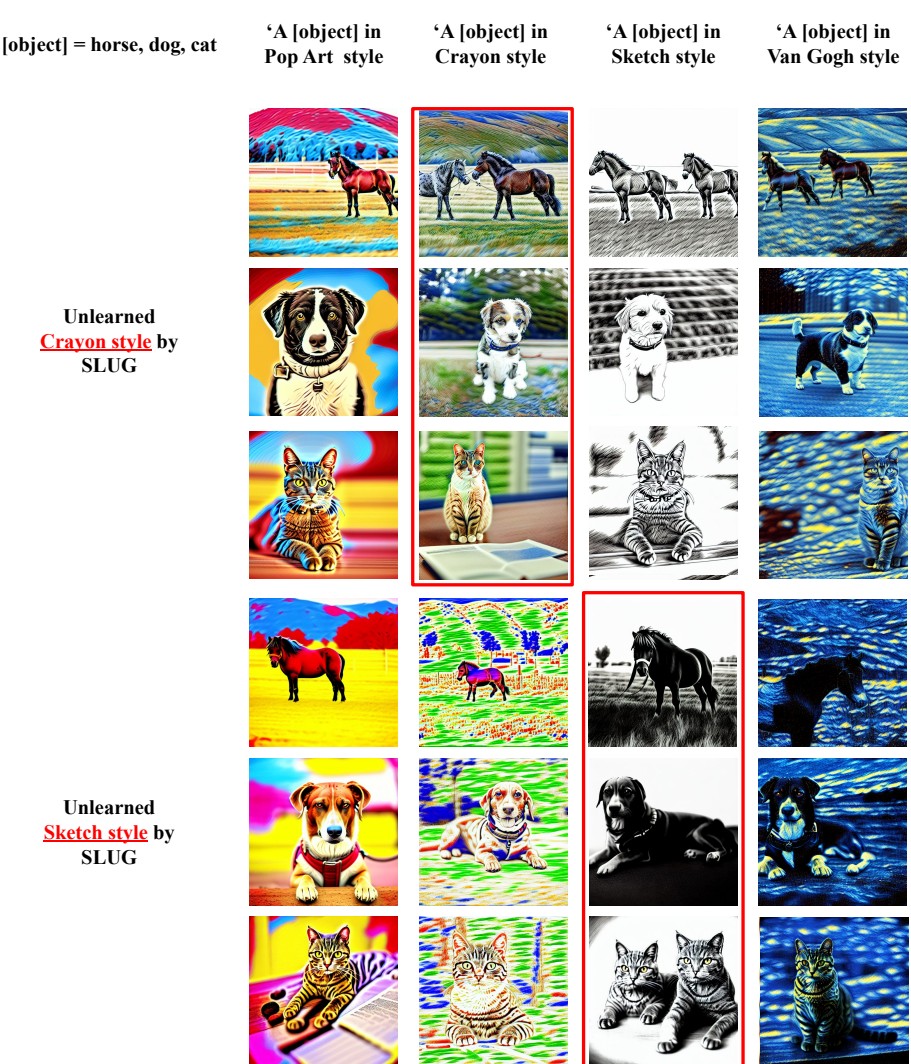

**Figure 19:** Visual examples of SLUG performance on UnlearnCanvas. Row $1 - 3$: outputs from UnlearnCanvas SD unlearned Crayon style. Row $4 - 6$: outputs from UnlearnCanvas SD unlearned Sketch style. Outputs corresponding to the unlearned style are highlighted by the red bounding box .

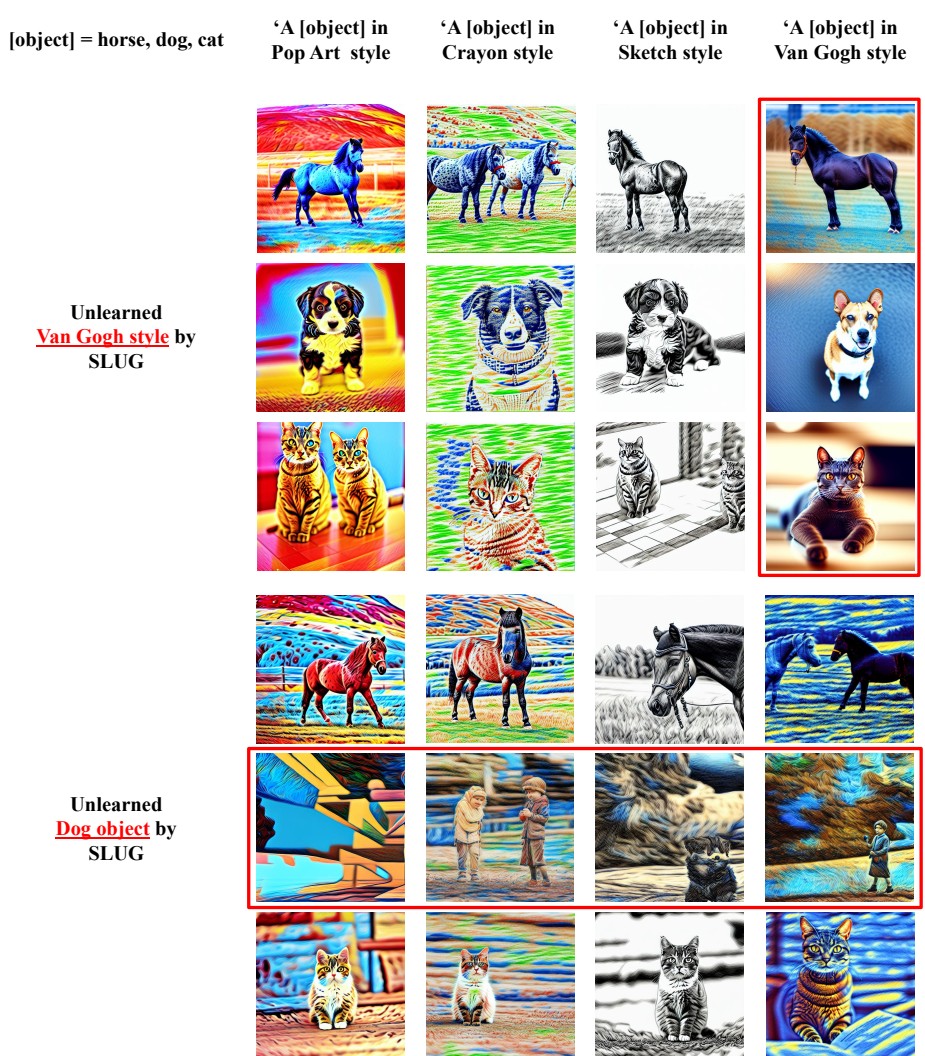

**Figure 20:** Visual examples of SLUG performance on UnlearnCanvas. Row $1-3$: outputs from UnlearnCanvas SD unlearned Van Gogh style. Row $4-6$: outputs from UnlearnCanvas SD unlearned dog object. Outputs corresponding to the unlearned style/object are highlighted by the red bounding box.

