# OpenReview forum: "Targeted Unlearning with Single Layer Unlearning Gradient"
_NeurIPS.cc/2024/Workshop/SafeGenAi — SafeGenAi Poster_

### Official Review · Reviewer_mLJB · 2024-10-08
**While the two metrics that allow targeting a specific layer are of great interest, the lack of an ablation regarding the one-time gradient computation, as well as the lack of a detailed description of the (fundamental) results described in Fig. 2 justifies the given rating of 5.**

**Rating:** 5
**Confidence:** 2

**Review:**

Summary:
The authors propose a novel unlearning strategy that identifies relevant layers for unlearning and only applies weight updates on these layers. The gradient is further only computed once, further reducing the computational cost of the unlearning scheme. They show that they obtain results competitive with alternative known approaches on Stable Diffusion when only training the language model behind CLIP.

Strengths:
 - Well written introduction to unlearning
 - The proposed linear scheme is computationally cheap as the gradients are only computed once
 - Targeting the concept relevant layers should help preserve the model capabilities, the fact that these layers are not chosen ad-hoc as typically done (attention layers) is especially interesting. The two used metrics are logical and general.

Weaknesses:
 - The figures outlining the results (such as Fig 2. or 7. or 13.) are not explained clearly enough. It is not immediately clear what should be interpreted. I would advise the authors to spend more time explaining what is shown in the text and what the reader can learn from it.
 - I would have liked to see which layers are most targeted, probably that information is in Fig 2. but writing it more explicitly would definitely give added value. Does this correspond with the expected trends in the literature?
 - The one-time gradient computation is surprising considering modern approaches. An ablation using typical SGD trained with gradients computed at each step is, in my opinion, necessary. If the authors have a good reason to believe that a linear approach is well grounded (beyond the computational cost) it should be mentioned more clearly.
 - How the step size $\lambda(t)$ varies (or it's link with the typical learning rate mentioned for other approaches) is not clear for to me, even though its role is fundamental to the approach.

Summary review:
While the two metrics that allow targeting a specific layer are of great interest, the lack of an ablation regarding the one-time gradient computation, as well as the lack of a detailed description of the (fundamental) results described in Fig. 2 justifies the given rating of 5.

---

### Official Review · Reviewer_6E4q · 2024-10-09
**An efficient unlearning method with gradient balance on a single-layer level**

**Rating:** 7
**Confidence:** 4

**Review:**

**Strengths**

- The method is efficient with only one-time gradient computation needed.
- The method offers a novel perspective on the trade-off between unlearning and retaining that considers layer importance and gradient alignment.
- The experiments demonstrate the effectiveness of unlearning and preservation of model utility.

**Weaknesses**

- Quantitative results of all unlearned identities should be included in the main paper, using only unlearning 'Elon Musk' as an example is less convincing.
- Previous works, such as SalUn and SSD, have explored localizing importance at a more fine-grained parameter level. The paper should empirically clarify why they focus on layer level.
- Since the paper identifies the layer with both importance on forgetting set and alignment with retaining set, it would be interesting to explore the Pareto optimal when unlearning multiple conflicting concepts.

---

### Official Review · Reviewer_aTAk · 2024-10-10
**A machine unlearning method SLUG**

**Rating:** 6
**Confidence:** 4

**Review:**

This paper proposes a novel machine unlearning method named SLUG that focuses on efficiently unlearning targeted information by updating a single layer in a model. This approach is distinct from more traditional methods that may require extensive retraining or fine-tuning of multiple layers, thereby offering a potential reduction in computational costs and complexity.

A major strength of this paper is its innovative approach to machine unlearning, which simplifies the process significantly by isolating changes to a single layer. This method not only reduces computational overhead but also limits the potential degradation of model performance on unrelated tasks—a critical consideration in practical applications.

However, a potential weakness of the study is the limited evaluation regarding the diversity of machine unlearning algorithms. The paper focuses exclusively on the SLUG method without a comparative analysis against a broad spectrum of existing unlearning approaches. This narrower focus might limit understanding of how SLUG performs relative to other state-of-the-art unlearning methods in various scenarios. Another limitation is that the experiments are confined to a single type of architecture—primarily Vision Transformers (ViTs). It remains uncertain whether the method’s reliance on layer importance metrics will be as effective on other architectures, such as Convolutional Neural Networks (CNNs), which have fundamentally different structural characteristics and learning behaviors. Given the distinct operational dynamics of CNNs compared to ViTs, the transferability and efficacy of SLUG across these differing architectures warrant further investigation to enhance its applicability and robustness.